# Plasma Interleukin-6 Level Predicts the Risk of Arteriovenous Fistula Dysfunction in Patients Undergoing Maintenance Hemodialysis

**DOI:** 10.3390/jpm13010151

**Published:** 2023-01-12

**Authors:** Jihyun Baek, Hyeyeon Lee, Taeyoung Yang, So-Young Lee, Yang Gyun Kim, Jin Sug Kim, ShinYoung Ahn, Kipyo Kim, Seok Hui Kang, Min-Jeong Lee, Dong-Young Lee, Hye Yun Jeong, Yu Ho Lee

**Affiliations:** 1Division of Nephrology, Department of Internal Medicine, CHA Bundang Medical Center, CHA University School of Medicine, Seongnam 13496, Republic of Korea; 2Division of Nephrology, Department of Internal Medicine, Kyung Hee University Hospital at Gangdong, College of Medicine, Kyung Hee University, Seoul 05278, Republic of Korea; 3Division of Nephrology, Department of Internal Medicine, College of Medicine, Kyung Hee University, Seoul 05278, Republic of Korea; 4Division of Nephrology, Department of Internal Medicine, College of Medicine, Korea University, Seoul 02841, Republic of Korea; 5Division of Nephrology and Hypertension, Department of Internal Medicine, Inha University College of Medicine, Incheon 22212, Republic of Korea; 6Division of Nephrology, Department of Internal Medicine, College of Medicine, Yeungnam University, Daegu 42415, Republic of Korea; 7Department of Nephrology, Ajou University School of Medicine, Suwon 16499, Republic of Korea; 8Division of Nephrology, Department of Internal Medicine, Veterans Health Service Medical Center, Seoul 03080, Republic of Korea

**Keywords:** arteriovenous fistula, hemodialysis, interleukin-6, vascular access

## Abstract

Systemic inflammation has been proposed as a relevant factor of vascular remodeling and dysfunction. We aimed to identify circulating inflammatory biomarkers that could predict future arteriovenous fistula (AVF) dysfunction in patients undergoing hemodialysis. A total of 282 hemodialysis patients were enrolled in this prospective multicenter cohort study. Plasma cytokine levels were measured at the time of data collection. The primary outcome was the occurrence of AVF stenosis and/or thrombosis requiring percutaneous transluminal angioplasty or surgery within the first year of enrollment. AVF dysfunction occurred in 38 (13.5%) patients during the study period. Plasma interleukin-6 (IL-6) levels were significantly higher in patients with AVF dysfunction than those without. Diabetes mellitus, low systolic blood pressure, and statin use were also associated with AVF dysfunction. The cumulative event rate of AVF dysfunction was the highest in IL-6 tertile 3 (*p* = 0.05), and patients in tertile 3 were independently associated with an increased risk of AVF dysfunction after multivariable adjustments (adjusted hazard ratio = 3.06, *p* = 0.015). In conclusion, circulating IL-6 levels are positively associated with the occurrence of incident AVF dysfunction in hemodialysis patients. Our data suggest that IL-6 may help clinicians identify those at high risk of impending AVF failure.

## 1. Introduction

The global incidence of end-stage kidney disease (ESKD) has been increasing over the past decades, and hemodialysis is the most commonly used treatment modality for patients with ESKD in Korea [1,2]. As vascular access is “the lifeline” for these patients, maintaining adequate vascular access patency is crucial [3,4]. In addition, the cost of managing dysfunctional vascular access is a major financial burden, representing a high percentage of the total cost related to the care of patients undergoing hemodialysis [5,6]. In this regard, adequate monitoring and prevention efforts for vascular access dysfunction are critical. Although the pathophysiology of vascular failure is not fully understood, previous studies have suggested that various factors such as old age, underlying diabetes, smoking, dyslipidemia, uremia, shear stress, and hypoxic injury are associated with poor vascular access patency [7,8,9,10]. Among these factors, the appreciation of the importance of inflammation has increased because neointimal hyperplasia is the most common process occurring in vascular dysfunction, and this process is linked to the inflammatory response [11].

Few studies have investigated biomarkers related to arteriovenous fistula (AVF) failure, particularly inflammatory markers. Discovering biomarkers is very important for understanding the pathophysiology of AVF failure and establishing future preventative therapeutic targets [11]. In this context, we investigated biomarkers associated with AVF failure including inflammation-related markers.

## 2. Materials and Methods

### 2.1. Study Population

An overview of the study design and patient recruitment strategy is shown in Figure 1. This retrospective multicenter cohort study was conducted using the K-cohort (CRIS No. KCT0003281) database, which enrolled 500 patients undergoing maintenance hemodialysis between June 2016 and April 2019 in six hospitals in South Korea. The inclusion criteria were age > 18 years and treatment with thrice-weekly hemodialysis for at least 3 months. Patients with a history of active infection, coagulation disorders, cancer, or kidney transplantation were excluded. Among the initially screened patients, 218 were excluded for the following reasons: withdrawal of informed consent (*n* = 6); lack of information regarding vascular access (*n* = 3); arteriovenous graft (AVG) (*n* = 71); central catheter (*n* = 17); lack of repository samples (*n* = 121). The remaining 282 patients with AVF were included in this study for analysis.

### 2.2. Data Collection

The baseline demographics and clinical parameters of the patients were obtained at the time of enrollment. Laboratory data obtained included plasma hemoglobin, albumin, calcium, phosphorus, intact parathyroid hormone, low-density lipoprotein cholesterol (LDL-C), erythrocyte sedimentation rate, and high-sensitivity C-reactive protein (hs-CRP). The Charlson comorbidity index was used as an indicator of the general health status of the patients [12]. Consent was obtained from all the patients.

### 2.3. Outcome Measures

The primary endpoint was the occurrence of AVF dysfunction within one year of enrollment. AVF dysfunction was defined as the development of arteriovenous fistula stenosis and/or thrombosis, requiring percutaneous transluminal angioplasty or surgery to revise or replace the fistula.

### 2.4. Measurements of Circulating Inflammatory Cytokines and Chemokines

Plasma samples were collected, processed, and stored as described in the previous study [13,14,15,16,17]. Magnetic Luminex^®^ Screening Assay multiplex kits (R&D Systems, Inc., Minneapolis, MN, USA) were used to conduct an enzyme-linked immunosorbent assay. Plasma samples for the measurement of plasma interleukin-6 (IL-6), monocyte chemoattractant protein-1 (MCP-1), and tumor necrosis factor-α (TNF-α) were collected using ethylenediaminetetraacetic acid-treated tubes at the time of study entry. After centrifugation for 15 min at 1000× *g* at room temperature, samples were stored at −80 °C until use. 

### 2.5. Statistical Analysis

Continuous variables are expressed as mean ± standard deviation or median (interquartile range (IQR)). Categorical variables are presented as numbers and percentages. Continuous variables were compared using the Student’s t-test or analysis of variance (ANOVA) with the Tukey post-hoc test. Non-normally distributed variables were compared by the Mann–Whitney U test or Kruskal–Wallis test. The chi-square (χ^2^) test or Fisher’s exact test was used to compare the categorical variables. Pearson’s correlation analysis was used to compare plasma IL-6 levels with hs-CRP and dialysis vintages. A binary logistic regression analysis was used to examine the risk factors for AVF failure. A multivariable analysis was performed with variables exhibiting a *p*-value of less than 0.1 in the univariable analysis. The intervention-free survival of patients with AVF was analyzed using the Kaplan–Meier curve and log-rank test. Univariable and multivariable analyses to investigate the associations between various parameters and AVF failure were performed using the Cox proportional hazards models. Parameters that were significantly associated with weight in the univariable analysis and clinically fundamental parameters were used as adjustment variables. The receiver operating characteristic (ROC) curve was generated, and areas under the curve (AUC) were calculated to evaluate the discriminative power of plasma IL-6 levels to identify those with AVF dysfunction. Statistical analyses were performed using the IBM SPSS Statistics for Windows (version 27; IBM Corp., Armonk, NY, USA). Statistical significance was set at *p* < 0.05.

## 3. Results

### 3.1. Baseline Clinical Parameters of Enrolled Patients

During the one-year follow-up period, 38 (13.5%) AVF dysfunctions occurred. The clinical characteristics and laboratory data of patients according to the occurrence of AVF dysfunction are shown in Table 1. The mean age and dialysis duration did not differ between the groups. Patients with AVF dysfunction showed significantly higher Charlson comorbidity index scores and marginally higher prevalence of diabetes, and they were more frequently prescribed statins than those without AVF dysfunction (4.3 ± 1.3 vs. 4.0 ± 1.6, 71.1% vs. 54.9%, and 65.8% vs. 43.4%; *p* = 0.019, *p* = 0.062, and *p* = 0.010, respectively). Vascular vintage was comparable between patients with and without AVF dysfunction (2.1 [0.6–3.9] vs. 2.2 [0.7–4.0] years, *p* = 0.656). Serum phosphorus and intact parathyroid hormone levels were significantly lower in the AVF dysfunction group than in the intact AVF group (4.36 ± 1.55 vs. 4.91 ± 1.34 mg/dL and 196.4 ± 158.1 vs. 290.6 ± 228.2 pg/dL, *p* = 0.023 and 0.015, respectively), while serum calcium levels were not different between groups. Among the measured proinflammatory markers, plasma IL-6 levels were significantly higher in patients with AVF dysfunction than in those without dysfunction (3.96 [2.58–7.56] vs. 2.76 [1.93–4.62] pg/mL, *p* = 0.009; Figure 2). There were no between-group differences in plasma hs-CRP, MCP-1, or TNF-α levels. Notably, there was a significant correlation between plasma IL-6 and hs-CRP levels (r = 0.406, *p* < 0.001; Appendix A). Plasma IL-6 levels were also correlated with the dialysis vintage; however, the correlation coefficient was weak (r = 0.133, *p* = 0.026).

We also compared the baseline clinical parameters of the enrolled patients according to the tertiles of plasma IL-6 levels (tertile 1, <2.228 pg/mL; tertile 2, ≥2.150 and <4.010 pg/mL; tertile 3, ≥4.010 pg/mL; Appendix A). Patients in tertile 3 were significantly older and showed higher levels of circulating inflammatory markers, including erythrocyte sedimentation rate, hs-CRP, and TNF-α, than those in tertile 1.

### 3.2. Risk Factors for the Occurrence of AVF Dysfunction 

The univariable logistic regression analysis revealed that diabetes, high comorbidity index, low systolic blood pressure, statin use, and low serum phosphorus levels were associated with the occurrence of AVF dysfunction within one year (Table 2). Those in tertile 3 showed an increased risk of AVF dysfunction compared to those in tertile 1 (odds ratio (OR) = 3.12, 95% confidence interval (CI) = 1.24–7.87, *p* = 0.016). The multivariable logistic regression analysis further demonstrated that the presence of diabetes and low systolic blood pressure were independently associated with AVF dysfunction (adjusted OR = 2.20, 95% CI = 1.00–4.86, *p* = 0.049 and adjusted OR = 1.34, 95% CI = 1.10–1.19 per 10 mmHg decrease in systolic blood pressure, *p* = 0.003, respectively). Plasma IL-6 levels also remained a relevant risk factor for the occurrence of AVF dysfunction (adjusted OR = 3.58, 95% CI = 1.36–9.40, *p* = 0.01).

### 3.3. Association between Plasma IL-6 Level and AVF Dysfunction 

The Kaplan–Meier curve showed that IL-6 tertile 3 had the highest cumulative incidence rate of AVF dysfunction (*p* = 0.05; Figure 3). Consistently, the univariable Cox regression analysis revealed that IL-6 tertile 3 was significantly associated with an increased risk of AVF dysfunction in comparison with IL-6 tertile 1 (hazard ratio [HR] = 2.86, 95% CI = 1.20–6.86, *p* = 0.018) (Table 3). This association remained significant after adjusting for multiple variables (adjusted HR = 3.06, 95% CI = 1.25–7.49, *p* = 0.015).

The relationship between plasma IL-6 and AVF dysfunction events was further investigated in predefined subgroups stratified by the presence of diabetes mellitus (DM) and IL-6 levels, with a cut-off value defined as a median value (>2.945 pg/mL; Table 4). The highest cumulative event rate was observed in patients with diabetes with high IL-6 levels (20.3%). A multivariable analysis showed that high IL-6 levels were independently associated with an increased risk of AVF dysfunction in both patients with and without diabetes (adjusted HR = 4.81, 95% CI = 1.02–22.64, *p* = 0.047 and HR = 7.53, 95% CI = 1.49–38.15, *p* = 0.015, respectively). Patients with diabetes with low IL-6 levels showed a marginally increased risk of AVF dysfunction (HR = 4.36, 95% CI = 0.87–21.94, *p* = 0.074). No significant interaction was observed between diabetes and plasma IL-6 levels (*p* for interaction = 0.953).

### 3.4. Discriminative Power of IL-6 to Identify AVF Dysfunction within One Year

Finally, a ROC curve analysis was performed to evaluate the diagnostic power of plasma IL-6 levels in identifying patients at high risk of future AVF dysfunction (Figure 4). The covariable-adjusted ROC curve showed the fair discriminative power of plasma IL-6, with an adjusted AUC of 0.73 (*p* < 0.001).

## 4. Discussion

In this study, we investigated the associations between circulating inflammatory cytokines and the occurrence of future AVF dysfunction in patients undergoing maintenance hemodialysis. The principal finding of this study was that patients in IL-6 tertile 3 had the greatest risk of AVF dysfunction, and that this association remained significant after adjusting for established clinical risk factors. Our findings suggest that plasma IL-6 may be a novel biomarker of the incident of AVF dysfunction in hemodialysis patients.

Among the complications associated with vascular access, AVF failure is the critical cause of morbidity in patients with ESKD undergoing hemodialysis [4]. Although clinical guidelines recommend the arteriovenous fistula as the first choice of vascular access due to lower complication incidence and relative long-term patency, AVFs are also exposed to patency loss [18]. Previous studies have investigated the relationship between vascular access dysfunction and mortality patients with ESKD and showed significant associations between these two variables [19,20]. A recent study also showed that early vascular access failure was an independent factor for a higher risk of mortality in hemodialysis patients [20], and another study suggested that recurrent vascular patency loss could be a predictive marker of all-cause mortality and cardiovascular events [19].

Various factors have been suggested as risk factors for vascular access dysfunction. Although the mechanism of AVF dysfunction has not been fully elucidated, neointimal hyperplasia is thought to be the most critical factor for the late loss of vascular access patency [21,22,23]. Neointimal hyperplasia in AVF results from multiple vascular biological pathways including inflammation, uremia, shear stress, and increased thrombogenicity commonly observed in hemodialysis patients [24]. These mechanisms are considered to act cooperatively through linked cytokine cascades and induce negative remodeling to occur. As a result, progressive neointimal hyperplasia gradually narrows the venous outflow and leads to stenosis, which can cause thrombosis of AVF. It is known that the vascular stenosis is histologically similar to atherosclerotic lesions, indicating intimal hyperplasia with active smooth muscle cell proliferation and extracellular matrix deposition [25,26]. Furthermore, increased levels of platelet-derived and mitogenic factors for smooth muscle cells were observed in both cases. In this context, it has been suggested that the AVF failure model might present a similar environment to accelerated atherosclerosis, which is characteristically observed in patients with ESKD [27].

Several studies have suggested that inflammation may contribute to atherogenesis. In addition to systemic inflammation caused by uremia, repeated cannulations in AVF result in a sustained local inflammatory response characterized by T cell and macrophage infiltrations and increased proinflammatory cytokines [28,29,30]. The importance of inflammation in AVF failure is further demonstrated in a study that showed significant decreases in intimal hyperplasia in MCP-1 knockout mouse models [31]. Among the various factors associated with the inflammatory process, IL-6 is known to be a key molecule in many studies. This molecule is a potent cytokine related to increased cardiovascular risk and atherosclerosis, not only in the general population but also in hemodialysis patients [32,33,34,35]. Previous studies have shown that inflammatory mechanisms associated with increased IL-6 levels might also affect vascular access stenosis and thrombosis. Chang et al. compared the inflammatory activity of thrombosed versus non-thrombosed AVF using venous segments to examine the expression of inflammation-associated molecules. They showed significantly higher expressions of IL-6 and TNF-α in the thrombosed group [36]. Dukkipati et al. showed that serum levels of inflammatory molecules including IL-6 were significantly higher in AVG compared with AVF, which is known to be a vascular access having a lower rate of dysfunction [37]. Our study also showed that higher IL-6 levels were significantly related to AVF failure, and this relationship was also significant in the multivariable Cox regression model.

As vascular access failure imposes a considerable cost on health care, and this affects not only the quality of life but also the survival outcome of dialysis patients, the efficient functioning of vascular access is a critical problem for these patients. Although therapeutic methods for vascular access failure have been relatively well-developed, strategies for preventing vascular access dysfunction are rare. To establish therapeutic and preventive targets, detecting circulating biomarkers is valuable for identifying the risk of AVF failure. However, few studies have assessed the relationship between AVF failure and circulating biomarker levels. In addition, there has been little consensus on the relationship between conflicting and reported results. Morton et al. recently performed a systemic analysis for this purpose, aiming to identify circulating biomarkers associated with AVF failure, but they could not identify valuable biomarkers, resulting in the inability of the currently assessed blood markers to identify the AVF failure risk [38]. In addition, most studies have evaluated biomarkers restricted to those used in routine clinical investigations, suggesting that more studies on more plausible biomarkers are necessary. Although the present study could not identify numerous biomarkers with a significant predictive value for the AVF failure risk, we used various biomarkers related to inflammation, which is known to be a significant mechanism of vascular access failure. Additional studies are necessary to establish AVF failure-related biomarkers to provide valuable directions for early detection and prevention of AVF dysfunction.

Several previous studies investigated the effects of statin use on AVF dysfunction in hemodialysis patients [39,40,41]. However, their association remains inconclusive. In our study, statin use was significantly associated with an increased risk of AVF dysfunction even after multiple adjustments including the presence of diabetes (Table 2). Nevertheless, it was difficult to confirm their relationship, given the presence of unrevealed differences in baseline characteristics and laboratory parameters between statin users and non-users. Well-designed randomized controlled trials are required to clarify the effects of statin on AVF failure. Diabetes is an important risk factor for AVF dysfunction [42,43]. In addition, diabetes causes a hyperglycemia-induced upregulation of proinflammatory cytokines including IL-6 [44]. In this regard, we performed a subgroup analysis of patients according to the presence of diabetes and plasma IL-6 levels. Our study showed that the association between IL-6 levels and AVF dysfunction events was not significant in a subgroup of DM patients with low IL-6 levels. In contrast, the risk of events increased in those without DM with high IL-6 levels, eventually showing the greatest risk of AVF dysfunction in patients with DM with high IL-6 levels. These findings would suggest that IL-6 could be a probable marker for screening patients at high risk for AVF dysfunction combined with DM.

This study had some limitations. Information on surveillance and the severity of AVF dysfunction was not obtained. Although venous pressure measurements are frequently used for this purpose, they are likely to be very limited in fistula surveillance [45]. Duplex ultrasound is an effective and widely used method for vascular access monitoring [46]; however, it is a labor-intensive and time-consuming method that relies on special equipment. Therefore, the surveillance of AVF based on circulating biomarkers may be a more cost-effective and convenient tool for the early detection and prevention of AVF failure. Second, we could not evaluate temporal changes in inflammatory marker levels over time. Third, we could not measure other parameters that might be valuable to evaluate vascular function or injury, such as D-dimer, the Von-Willebrand Factor, thrombomodulin, or plasminogen activator inhibitor-1. Finally, as this was a prospective observational cohort study, the event time of vascular access failure could not be calculated from the vascular access operation time. To compensate for this limitation, we assessed the AVF vintage to investigate its association with AVF dysfunction and found no significant relationship between the two variables.

## 5. Conclusions

We demonstrated that high plasma IL-6 levels were independently associated with an increased risk of incidents of AVF dysfunction in patients undergoing hemodialysis. These findings suggest that IL-6 may play a role in the progression of AVF remodeling and subsequent failure. Further investigations on a larger number of patients over a longer period are needed to prove the importance of the inflammatory parameter in AVF function assessment and to validate the diagnostic utility of plasma IL-6 for future AVF dysfunction. 

## Figures and Tables

**Figure 1 jpm-13-00151-f001:**
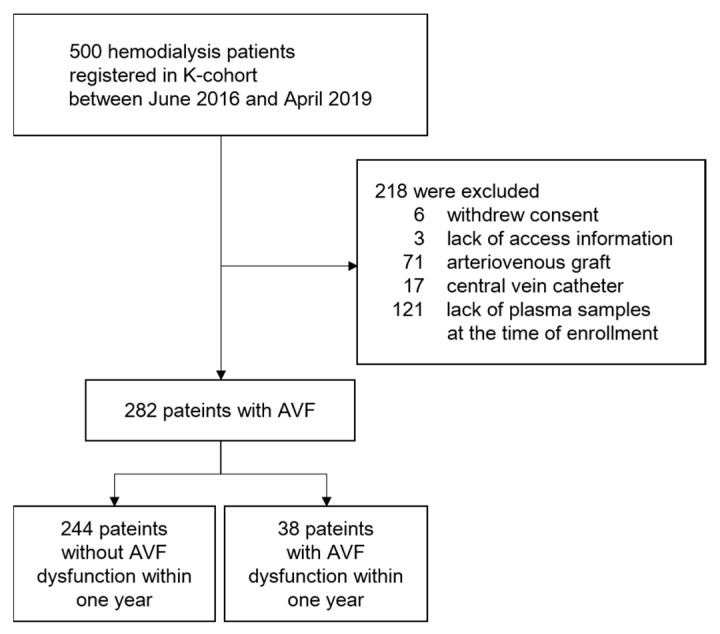
A flowchart of the study participant selection. Abbreviation: AVF, arteriovenous fistula.

**Figure 2 jpm-13-00151-f002:**
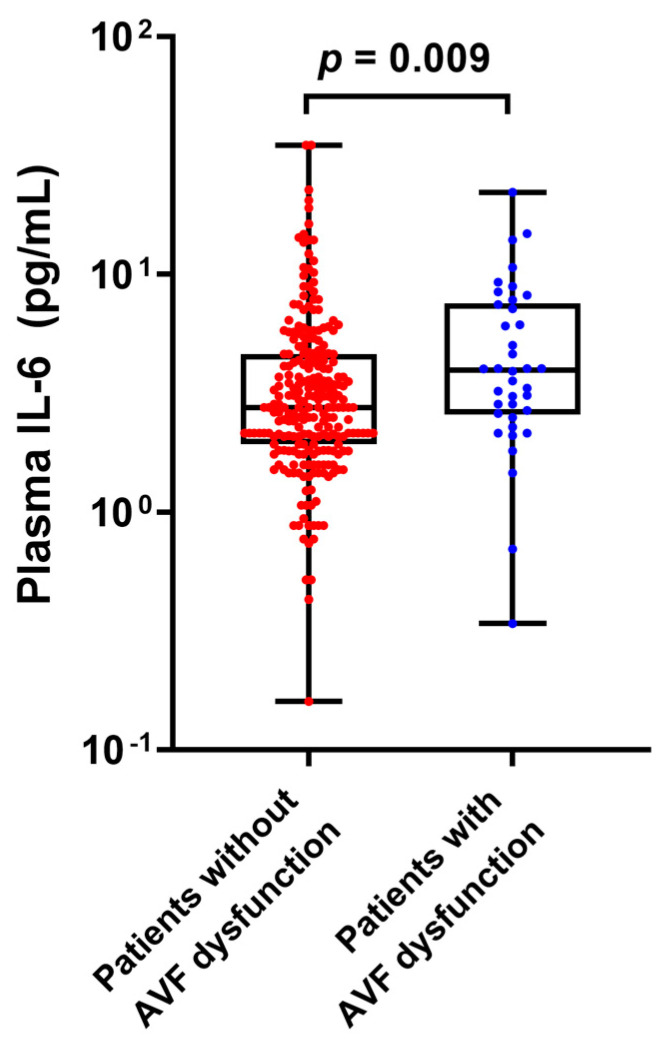
Plasma IL−6 levels according to the occurrence of AVF dysfunction.

**Figure 3 jpm-13-00151-f003:**
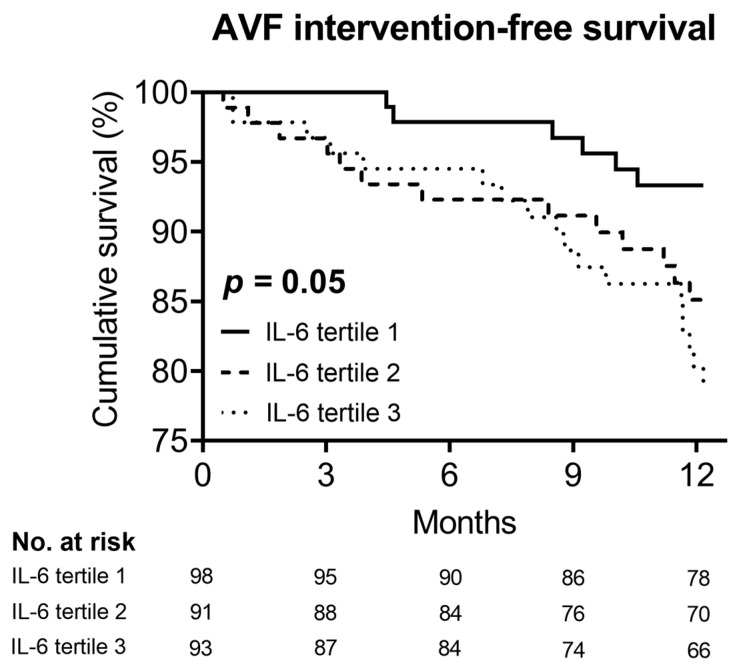
Cumulative survival rate of arteriovenous fistula according to plasma IL-6 tertiles. Abbreviations: AVF, arteriovenous fistula; IL-6, interleukin-6.

**Figure 4 jpm-13-00151-f004:**
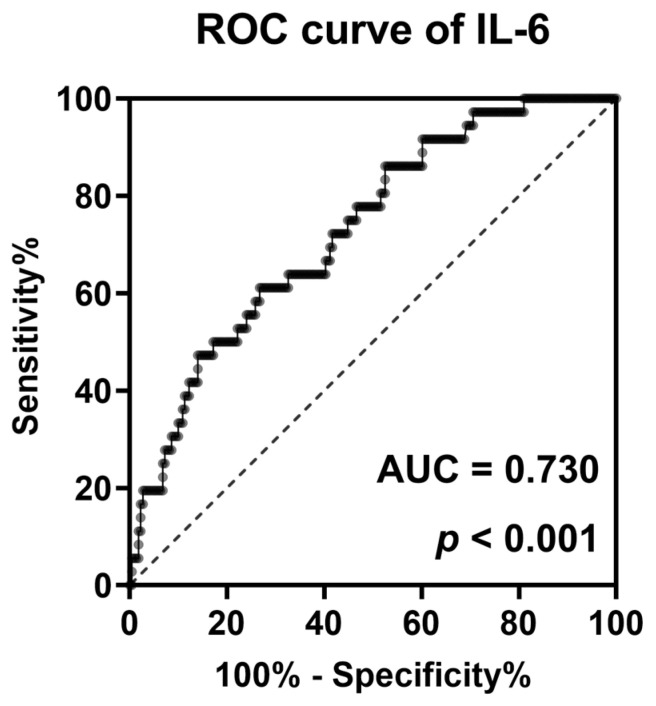
A covariable-adjusted ROC curve for the evaluation of the discriminative power of plasma IL-6 levels. Adjusted for the following covariables: age, sex, diabetes mellitus, arteriovenous fistula vintage, systolic blood pressure, and statin use. Abbreviations: ROC, receiver operating characteristic; AUC, area under the curve.

**Table 1 jpm-13-00151-t001:** Baseline demographic and laboratory data of the study population according to the occurrence of arteriovenous fistula dysfunction within one year after enrollment.

	Patients without	Patients with	*p* Value
	AVF Dysfunction	AVF Dysfunction	
	(*n* = 244)	(*n* = 38)	
Age (year)	60.7 ± 13.3	63.4 ± 11.3	0.162
Male (*n*, %)	162 (66.4)	28 (73.7)	0.373
Body mass index (kg/m^2^)	23.6 ± 5.5	23.7 ± 5.0	0.836
Dialysis vintage (year) ^a^	2.1 (0.7, 4.9)	1.7 (0.6, 3.8)	0.203
Charlson comorbidity index	4.0 ± 1.6	4.3 ± 1.3	0.019
Diabetes mellitus (*n*, %)	134 (54.9)	27 (71.1)	0.062
Previous history of CVD (*n*, %) ^b^	218 (89.3)	36 (94.7)	0.301
Pre-dialysis systolic BP (mmHg)	145.4 ± 20.0	136.8 ± 18.3	0.013
Location of AVF (*n*, %) ^c^			
Forearm	186 (76.5)	26 (68.4)	0.279
Upper arm	57 (23.5)	12 (31.6)	
AVF vintage (year) ^a^	2.21 (0.7–4.0)	2.08 (0.6–3.9)	0.656
Ultrafiltration (L/session)	2.27 ± 1.04	2.46 ± 1.00	0.310
Single pool Kt/V	1.59 ± 0.29	1.56 ± 0.28	0.561
Blood flow rate (mL/min)	269.0 ± 22.1	269.0 ± 28.6	0.964
Hemodialysis duration (h)	3.91 ± 0.21	3.94 ± 0.16	0.301
Hemodiafiltration (*n*, %)	56 (23.0)	14 (36.8)	0.065
Statin use (*n*, %)	106 (43.4)	25 (65.8)	0.010
Anti-platelet agent use (*n*, %)	173 (70.9)	28 (73.7)	0.724
Erythropoiesis-stimulating agent use (%)	226 (92.6)	32 (84.2)	0.084
Hemoglobin (g/dL)	10.4 ± 1.2	10.5 ± 1.4	0.834
Intact parathyroid hormone (pg/dL)	290.6 ± 228.2	196.4 ± 158.1	0.015
Calcium (mg/dL)	8.52 ± 0.83	8.51 ± 0.75	0.934
Phosphorus (mg/dL)	4.91 ± 1.34	4.36 ± 1.55	0.023
Albumin (mg/dL)	3.83 ± 0.30	3.77 ± 0.36	0.231
Alkaline phosphatase (mg/dL)	109.1 ± 78.9	104.3 ± 57.5	0.926
LDL-cholesterol (mg/dL)	77.1 ± 24.6	76.2 ± 31.0	0.842
Erythrocyte sedimentation rate (mm/h)	30.4 ± 22.7	33.0 ± 23.1	0.630
hs-CRP (mg/dL) ^a^	0.80 (0.17–3.04)	0.90 (0.45–3.13)	0.518
IL-6 (pg/mL) ^a^	2.76 (1.93–4.62)	3.96 (2.58–7.56)	0.009
MCP-1 (pg/mL) ^a^	167.26 (133.42–216.58)	154.25 (118.80–200.67)	0.149
TNF-α (pg/mL) ^a^	9.83 (6.36–13.23)	10.15 (5.83–13.70)	0.962

Abbreviations: AVF, arteriovenous fistula; CVD, cardiovascular disease; BP, blood pressure; LDL, low-density lipoprotein; hsCRP, high-sensitivity C-reactive protein; IL-6, interleukin-6; MCP-1, monocyte chemoattractant protein-1; TNF-α, tumor necrosis factor-α. ^a^ Data are expressed as median (first and third interquartile ranges) and are compared by the Mann–Whitney test because of their non-normal distributions. ^b^ Includes congestive heart failure, myocardial infarction, coronary artery disease requiring percutaneous transluminal coronary angioplasty or coronary artery bypass surgery, ventricular arrhythmia, cardiac arrest, and sudden death. ^c^ No information in a patient without AVF dysfunction.

**Table 2 jpm-13-00151-t002:** Multivariable analysis for the prediction of arteriovenous fistula dysfunction.

		Univariable Analysis	Multivariable Analysis		
		OR (95% CI)	*p* Value	OR (95% CI)	*p* Value
Age (year)		1.02 (0.99–1.04)	0.232		
Sex (male)		1.42 (0.66–3.06)	0.374		
Diabetes mellitus		2.02 (0.96–4.25)	0.065	2.20 (1.00–4.86)	0.049
Charlson comorbidity index		1.15 (0.93–1.43)	0.192		
AVF vintage (year)		0.97 (0.87–1.08)	0.585		
Systolic BP (per 10 mmg decrease)		1.27 (1.06–1.52)	0.01	1.34 (1.10–1.19)	0.003
Anti-platelet agent use		1.15 (0.53–2.50)	0.725		
Statin use		2.50 (1.22–5.13)	0.012	2.76 (1.28–5.93)	0.01
Albumin (mg/dL)		0.52 (0.18–1.51)	0.231		
Calcium (> 9.5 mg/dL)		0.71 (0.23–2.75)	0.706		
Phosphorus (> 5.5 mg/dL)		0.73 (0.19–1.07)	0.073	0.48 (0.20–1.19)	0.113
hs-CRP (log)		1.05 (0.88–1.25)	0.6		
IL-6 (pg/mL)	tertile1	Ref	-	Ref	-
	tertile2	2.17 (0.82–5.70)	0.117	2.34 (0.85–6.44)	0.099
	tertile3	3.12 (1.24–7.87)	0.016	3.58 (1.36–9.40)	0.01

Multivariable analysis was performed with variables exhibiting *p* < 0.1 in univariable analysis. Abbreviations: OR, odds ratio; CI, confidence interval; AVF, arteriovenous fistula; BP, blood pressure; LDL, low-density lipoprotein; hs-CRP, high-sensitivity C-reactive protein; IL-6, interleukin-6.

**Table 3 jpm-13-00151-t003:** Hazard ratios of plasma IL-6 tertiles for arteriovenous fistula dysfunction.

	No. of Events (%)	Crude HR (95% CI)	*p*-Value	Adjusted HR (95% CI) ^a^	*p*-Value
IL-6 Tertile 1	7/98 (7.1%)	Reference		Reference	
IL-6 Tertile 2	13/91 (14.3%)	2.04 (0.81–5.12)	0.128	1.99 (0.79–5.00)	0.144
IL-6 Tertile 3	18/93 (19.4%)	2.86 (1.20–6.86)	0.018	3.06 (1.25–7.49)	0.015

Abbreviations: HR, hazard ratio; CI, confidence interval. ^a^ Adjusted for the following covariables: age, sex, diabetes mellitus, arteriovenous fistula vintage, systolic blood pressure, and statin use.

**Table 4 jpm-13-00151-t004:** Hazard ratios of plasma IL-6 levels for the arteriovenous fistula dysfunction according to predefined subgroups.

	No. of Events (%)	Crude HR (95% CI)	*p*-Value	Adjusted HR (95% CI) ^a^	*p*-Value	*p* for Interaction
						0.953
Low IL-6, without DM	2/59 (3.4%)	Reference	-	Reference		
High IL-6, without DM	9/62 (14.5%)	4.49 (0.97–20.79)	0.055	4.81 (1.02–22.64)	0.047	
Low IL-6, with DM	11/82 (13.4%)	4.04 (0.90–18.24)	0.069	4.36 (0.87–21.94)	0.074	
High IL-6, with DM	16/79 (20.3%)	6.38 (1.47–27.76)	0.013	7.53 (1.49–38.15)	0.015	

Abbreviations: HR, hazard ratio; CI, confidence interval. High IL-6 was defined as >2.9450 pg/mL. ^a^ Adjusted for the following covariables: age, sex, Charlson comorbidity index, arteriovenous fistula vintage, systolic blood pressure, and statin use.

## Data Availability

The data that support the findings of this study are available from the corresponding author upon reasonable request.

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
