# Peer review of "Plasma Interleukin-6 Level Predicts the Risk of Arteriovenous Fistula Dysfunction in Patients Undergoing Maintenance Hemodialysis"

_jpm, 2023, doi:10.3390/jpm13010151_

Round 1

Reviewer 1 Report

In this prospective epidemiological study, Korean authors wisely analyze factors contributing to native fistula dysfunction and elegantly show its association with high serum IL6 level and diabetes mellitus. 

Reviewer 2 Report

Interesting work. Perhaps it would be desirable in the conclusion to emphasize that the results should be confirmed on a larger number of patients over a longer period in order to prove the importance of inflammatory parameters in the assessment of AV fistula function.

Reviewer 3 Report

Authors investigated biomarkers associated with AVF failure including inflammation related markers.

Although this manuscript is potentially interesting, several issues arise.

1.     What is the mechanism of the onset of AVF dysfunction? Please discuss in Discussion.

2.     Was the IL-6 levels were correlated with CRP?

3.     What did increase IL-6 levels?

4.     Plasma levels of D-dimer, VWF, TM or PAI-I may be helpful.

5.     Is conclusion true? Is IL-6 level useful to predict AVF dysfunction?

6.     IL-6 may be useful for diagnosing AVF dysfunction. ROC analysis may be helpful.

7.     There was not big difference of IL-6 levels between with and without AVF dysfunction. 

8.     Relationship between hemodialysis period and IL-6 levels may be helpful.

9.     How is the severity of AVF dysfunction evaluated?

Round 2

Reviewer 3 Report

Authors sufficiently responded reviewer’s comments.

I have no further comment.